

# Effects of the acanthocephalan *Polymorphus minutus* and the microsporidian *Dictyocoela duebenum* on energy reserves and stress response of cadmium exposed *Gammarus fossarum*

Hui-Yu Chen[1,2,*], Daniel S. Grabner[2,*], Milen Nachev[2], Hsiu-Hui Shih[1] and Bernd Sures[2,3]

[1] Department of Life Science, National Taiwan University, Taipei, Taiwan
[2] Aquatic Ecology and Centre for Water and Environmental Research, University of Duisburg–Essen, Essen, Germany
[3] Department of Zoology, University of Johannesburg, Johannesburg, South Africa
* These authors contributed equally to this work.

Corresponding author
Daniel S. Grabner,
daniel.grabner@uni-due.de

## ABSTRACT

Amphipods are commonly parasitized by acanthocephalans and microsporidians and co-infections are found frequently. Both groups of parasites are known to have severe effects on their host. For example, microsporidians can modify host sex ratio and acanthocephalans can manipulate the behavior of the amphipod to promote transmission to the final host. These effects influence host metabolism in general and will also affect the ability of amphipods to cope with additional stressors such as environmental pollution, e.g., by toxic metals. Here we tested the effects of sub-lethal concentrations of cadmium on glycogen and lipid levels, as well as on the 70kDa heat shock protein (hsp70) response of field collected *Gammarus fossarum*, which were naturally infected with microsporidians and the acanthocephalan *Polymorphus minutus*. Infected and uninfected *G. fossarum* were exposed to a nominal Cd concentration of 4 µg/L, which resembled measured aqueous Cd concentration of 2.9 µg/L in reconstituted water for 7 d at 15 °C in parallel to an unexposed control. After exposure gammarids were snap frozen, weighed, sexed and tested for microsporidian infection by PCR. Only individuals containing the microsporidian *Dictyocoela duebenum* were used for the further biochemical and metal analyses. *P. minutus* infected amphipods were significantly smaller than their uninfected conspecifics. Mortality was insignificantly increased due to cadmium exposure, but not due to parasite infection. Microsporidian infection in combination with cadmium exposure led to increased glycogen levels in female gammarids. An increase of glycogen was also found due to interaction of acanthocephalan and microsporidian infection. Elevated lipid levels were observed in all groups infected with microsporidians, while acanthocephalans had the opposite effect. A positive correlation of lipid and glycogen levels was observed. The general stress response measured in form of hsp70 was significantly increased in microsporidian infected gammarids exposed to cadmium. *P. minutus* did not affect the stress response of its host. Lipid levels were correlated negatively with hsp70 response, and indicated a

possible increased stress susceptibility of individuals with depleted energy reserves. The results of our study clearly demonstrate the importance of parasitic infections, especially of microsporidians, for ecotoxicological research.

## INTRODUCTION

Parasites are natural and ubiquitous parts of ecosystems and their role for ecosystem function and stability has been recognized in recent years (*Marcogliese, 2005*; *Hudson, Dobson & Lafferty, 2006*; *Kuris et al., 2008*). Nevertheless, parasites are often neglected in ecotoxicological research, although results of recent studies showed that they may influence the host response to pollution (*Sures, 2004*; *Sures, 2008a*; *Sures, 2008b*; *Marcogliese & Giamberini, 2013*) and can impact the stress response of their host, especially on the biochemical level (*Sures, Lutz & Kloas, 2006*; *Sures & Radszuweit, 2007*; *Frank et al., 2011*; *Frank et al., 2013*; *Gismondi, Cossu-Leguille & Beisel, 2012a*; *Gismondi et al., 2012a*; *Gismondi et al., 2012b*; *Filipović Marijić, Vardić Smrzlić & Raspor, 2013*; *Grabner, Schertzinger & Sures, 2014*). Such impacts are particularly relevant when using stress responses on the biochemical level as biomarkers in environmental monitoring programs to indicate the presence of contaminants and to unravel their effects on organisms. As biomarkers are applied under field conditions, their possible modulation by naturally occurring stressors such as parasites has to be evaluated preferably under laboratory conditions as the knowledge about possible interactions between parasites and pollution is still limited (*Sures, 2008a*; *Marcogliese & Pietrock, 2011*).

Amphipods are important elements of aquatic ecosystems due to their widespread distribution, high biomass, and their ecological function as shredders of organic material (*Schirling et al., 2005*; *Platvoet et al., 2006*; *Navel et al., 2010*; *Piscart et al., 2011*). Because of their sensitivity to contaminants, they are often used as test organism in ecotoxicological studies and particularly in metal toxicity tests (*Schill, Görlitz & Köhler, 2003*; *Sures & Radszuweit, 2007*; *Lebrun et al., 2015*). Amphipods are also affected by several parasite groups including acanthocephalans and microsporidians (*Taraschewski, 2000*; *Stentiford et al., 2013*).

Acanthocephalans are a group of unique helminth parasites that have an invertebrate intermediate and a vertebrate final host (*Taraschewski, 2000*). They often grow to a large size in the body cavity of their amphipod intermediate hosts and can impact them in various ways. For example, they can influence the activity of the host and its predator avoidance behavior (*Bauer et al., 2000*; *MacNeil et al., 2003a*; *MacNeil et al., 2003b*; *Benesh et al., 2008*) or affect its reproduction rates (*Ward, 1986*; *Zohar & Holmes, 1998*; *Bollache, Gambade & Cézilly, 2001*). Furthermore, acanthocephalans can influence the energy

metabolism of the host and its vulnerability to metals (*Gismondi, Cossu-Leguille & Beisel, 2012a*; *Gismondi, Cossu-Leguille & Beisel, 2012b*; *Gismondi, Beisel & Cossu-Leguille, 2012b*).

Microsporidians are a diverse group of endoparasitic fungi of invertebrates and vertebrates (*Smith, 2009*; *Stentiford et al., 2013*). Their transmission can be horizontal from one individual to another and/or vertical, directly from mother to offspring (*Dunn & Smith, 2001*). High prevalences and co-infection cases with up to three microsporidian species underline the importance of these parasites for amphipods (*Haine et al., 2004*; *Grabner, Schertzinger & Sures, 2014*; *Grabner et al., 2015*). Due to intense proliferation of sporogonic stages microsporidians can modulate the host metabolism or its hormone homeostasis leading to feminization of male individuals (*Bandi et al., 2001*; *MacNeil et al., 2003a*; *Smith, 2009*). Both effects are likely to influence the host response to toxic stressors compared to uninfected individuals. Even under asymptomatic conditions, microsporidians were found to influence the stress response and energy reserves of their amphipod host, when an additional chemical stressor was present (*Gismondi et al., 2012a*).

Cadmium is released to the environment by various human activities and is frequently found in freshwater ecosystems in concentrations up to several µg/L. It can affect aquatic organisms and whole communities (*EPA, 2001*). The combination of exposure to metals like cadmium and simultaneous parasite infections can influence the immune response of the host (*Cornet et al., 2009*) and can increase energy consumption for maintenance of the basic metabolic functions (*Gismondi et al., 2012b*). Both glycogen and lipids are important energy reserves in animals and their levels can be used as biomarkers for metabolic stress (*Plaistow, Troussard & Cézilly, 2001*). Additionally, heat shock proteins (hsp) can be used as general stress markers, suitable to detect effects of parasites and metal exposure in amphipods (*Schill, Görlitz & Köhler, 2003*; *Frank et al., 2013*). The effect of cadmium exposure in combination with infection with the acanthocephalan *Polymorphus minutus* and microsporidians was already investigated in detail for the amphipod *Gammarus roeseli* (*Gismondi et al., 2012b*). Both *G. roeseli* and *G. fossarum* are suitable hosts for *P. minutus*, however it was found that they respond differently (e.g., on biochemical and behavioral level) to infection (*Bauer et al., 2000*; *Helluy, 2013*). We assumed that *G. fossarum* might also react differently to the combination of stressors such as metal pollution and parasite infections. Therefore, the aim of the present study was to investigate the combined effects of parasitism and cadmium pollution on ecotoxicological biomarkers such as energy metabolism (total lipid and glycogen content) and general stress response (hsp70) of *G. fossarum*. For this purpose, we conducted laboratory exposure experiments with amphipods, which were naturally infected with microsporidians and acanthocephalans.

## MATERIAL AND METHODS

### Sampling of gammarids and exposure design

Gammarids were collected by kick-sampling from a small stream (Rumbach) near Mülheim (N51°25′28.628′; E6°54′25.553″), NRW, Germany in March 2014. This site is characterized by a mixed population of *Gammarus pulex* and *Gammarus fossarum*, and high prevalence of the acanthocephalan *Polymorphus minutus* in both amphipod species

(unpublished results). After sorting the collected gammarids under a binocular a total of 382 *G. fossarum* was used for the experiment.

The gammarids were grouped by acanthocephalan infection (only amphipods infected with single cystacanths were used) and kept in the laboratory for 2 days at 15 °C in reconstituted water (7.36 µM NaCl, 231.39 µM KCl, 117.13 µM NaHCO$_3$, 707.05 µM, MgSO$_4 \cdot$ 7H$_2$O, 303.70 µM). Alder leaves from the sampling site were used as food source. The exposure experiment was conducted in six 2 L-tanks for control and cadmium-exposed groups, respectively. Each tank contained both acanthocephalan-uninfected *G. fossarum* ($n = 16–17$) and gammarids infected with *P. minutus* ($n = 14–16$). All animals were kept in reconstituted water and cadmium was added to the exposed groups to achieve a nominal concentration of 4 µg/L (diluted from a 1 g/L Cd standard, Cd(NO$_3$)$_2$ in 0.5 M HNO$_3$, Bernd Kraft GmbH). The duration of the exposure was 7 d at 15 °C in a climate chamber with 12/12 light/dark-cycle. This exposure time was considered to be long enough, as *Gismondi, Cossu-Leguille & Beisel (2012b)* found effects of cadmium on energy reserves of gammarids already after 96h exposure. Experimental tanks were aerated to saturation. No food was provided during the experimental exposure. Water had a temperature of 14.5 °C ($\pm$0.5), a pH of 8.1 ($\pm$0.1) and a conductivity of 1010.1 µS/cm ($\pm$7.8) in the control tanks and 14.3 °C ($\pm$0.5), pH 8.0 ($\pm$0.2) and 1,015.1 µS/cm ($\pm$17.2) in the cadmium tanks. Water samples for metal analysis were taken from each tank at different time points: once before cadmium was added and after 5, 10, 20, 30, 60, 120, 240 min, followed by daily sampling until the end of the experiment. A sample of 5 mL was filtered (cellulose nitrate filter, pore size 0.45 µm; Sartorius AG, Göttingen, Germany), acidified with 5 µL HNO$_3$ (65% suprapure; Merck, Kenilworth, New Jersey, USA) and stored at $-20$ °C until metal analysis. If present, dead gammarids were removed from the tanks twice daily and were discarded. At the end of the exposure period, gammarids were removed from the aquaria with a sieve, blotted dry and weighed. Cystacanths were dissected from infected amphipods and stored in separate tubes. All samples were snap frozen individually in 1.5 mL reaction tubes in liquid nitrogen and stored at $-80$ °C.

## Metal analyses

Cadmium concentrations in water samples, gammarids and cystacanths were determined by electrothermal atomic absorption spectrometry (ET-AAS) as described in *Sures, Taraschewski & Haug (1995)*. Water samples from 2 tanks were pooled resulting in three samples for each group. The acidified water samples were determined without any further pretreatment in duplicate. For tissue analysis, whole gammarids (not tested for microsporidian infection status) or cystacanths were pooled to achieve sample wet weight ranging from 55 to 80 mg for gammarids (2 males and 2 females with and without *P. minutus* from control and cadmium-treatment) or 11 mg for cystacanths (one pool of 69 for control and 66 for cadmium-treatment). Samples were digested in a mixture of 1.3 mL nitric acid (65% HNO$_3$, suprapure, Merck) and 2.5 mL hydrogen peroxide (30% H$_2$O$_2$, AppliChem) using a microwave digestion procedure described by *Zimmermann et al. (2001)*. To determine the detection limits, blanks were prepared without addition of

**Table 1 Primers.** Specific primers used to detect microsporidians.

| Primer | Sequence (5′–3′) | Amplicon length | Source |
|---|---|---|---|
| Mspec 505 | F: CAT CAA CTA ACT TTG GGA AAC TAA G<br>R: TGG CCT CCC ACA CAT TCC GAG TG | 1300 bp | (*Grabner, Schertzinger & Sures, 2014*) |
| Micro MH | F: GTA GAA CTG CGA TGA TTT AGT CTG<br>R: GCT ATA CCA TGT TCC CCA TTG | 433 bp | This study |
| Dict | F: GGG CGA TTT ATT TGT TCT CCT GT<br>R: TGA TTT CTC TTC CGC AAT ACC AAA TC | 680 bp | This study |

sample. Cadmium concentrations of samples were calculated by fitting linear regression lines to the points defined by spiked samples and the corresponding integrated peak areas.

## Molecular identification of microsporidians

After thawing, the sex of gammarids was determined according to male genital papillae or female oostegites. Acanthocephalan infected individuals were dissected, cystacanths removed and stored at $-20\,°C$ for metal analysis. Each *G. fossarum* specimen was homogenized with micropestles in 0.1 M sodium phosphate buffer with 0.1 M KCl. The buffer volume was adjusted to the threefold wet weight of every individual (e.g., 60 μL of homogenization buffer for 20 mg of wet weight tissue). Samples were centrifuged at 14.000 × g for 15 min at 4 °C. The supernatant was used for quantification of total protein concentration and analysis of glycogen, lipid and 70 kDa heat shock protein (hsp70) (see below). The remaining pellet was used for molecular diagnosis of microsporidian infections.

DNA was extracted from pellets with a JETQUICK Tissue DNA spin kit (Genomed) according to manufacturer's instructions. For molecular detection of microsporidian infections, the primers V1 5′-CAC CAG GTT GAT TCT GCC TGA C-3′ (*Zhu et al., 1993*) and Micro_rev 5′-GAG TCA AAT TAA GCC GCA CAA TCC AC-3′ (*Krebes et al., 2010*), amplifying a part of the small subunit ribosomal RNA gene (ss rDNA) of microsporidians, were used. For initial identification of microsporidian species infecting the gammarids, PCR bands of 10 randomly selected samples were gel purified with a JETQUICK PCR Product Purification Spin Kit (Genomed, St. Louis, Missouri, USA) according to manufacturer's instructions and were sent for sequencing (GATC, Konstanz, Germany). All obtained sequences were tested for matches in the GenBank by blast-search (http://blast.ncbi.nlm.nih.gov/Blast.cgi).

According to the obtained sequences, specific primers for each of the three microsporidian species detected were designed (see Table 1) to test the remaining positive samples. All PCR reactions contained 4 μL of 5× OneTaq Hot Start Buffer (New England Biolabs), 0.2 mM dNTP mix (New England Biolabs, Ipswich, Massachusetts, USA), 0.5 μM of each primer (MWG Eurofins), 0.5 U OneTaq Hot Start (New England Biolabs) and 1 μL template DNA. The mix was topped up to 20 μL with PCR grade water. The DNA was amplified by a Labcycler (SensoQuest, Göttingen, Germany). PCR conditions for the V1/Micro_rev primers were 94 °C for 3 min, 40 cycles of 94 °C for 30 s, annealing at 60 °C for 35 s, elongation at 68 °C for 1min and a final elongation at 68 °C for 3 min. The other primers were used under the following conditions: initial denaturation at 94 °C

for 5 min, 35 cycles of 94 °C for 30 s, annealing at 60 °C for 35 s and elongation at 68 °C for 40 s followed by a final elongation of 68 °C for 3 min. PCR bands were analyzed by conventional agarose gel electrophoresis. Specificity of diagnostic primers was assessed by sequencing of three randomly selected PCR products of each primer pair and comparison to the sequences obtained with the primers V1/Micro_rev.

## Biochemical analyses

### Total protein analysis

The amount of total protein in the samples supernatants was measured with a BCA Protein Assay Kit (Pierce, Waltham, Massachusetts, USA) according to manufacturer's instructions. A 10 µL aliquot of the supernatant was diluted 1:10 with homogenization buffer and 25 µL of the diluted samples were subsequently measured in triplicates in 96-well plates together with a dilution series of a bovine serum albumin standard in a plate reader (Tecan infinite M200).

### Glycogen analysis

The pellet obtained after the methanol/chloroform-extraction (see below) was dissolved in 200 µL of deionized water. Of this solution, 100 µL were transferred into clean 15 mL culture tubes with 4.9 mL of anthrone reagent (0.2 g of anthrone in 100 mL of 95% $H_2SO_4$). The mixture was heated in a water bath at 95 °C for 17 min, and was subsequently cooled on ice. The optical density was measured at 625 nm in 96-well plates in triplicates. As a reference, a standard of 25–800 µg/mL of glucose was used. Measurements were expressed as µg glycogen/µg total protein.

### Lipid analysis

To measure total lipid and glycogen content we used the method described by *Gismondi et al. (2012a)*, which was adapted from *Plaistow, Troussard & Cézilly (2001)*. After homogenization and centrifugation (see above), 10 µL of 2% sodium sulfate (w/v) and 270 µL of chloroform/methanol 1:2 (v/v) were added to 20 µL of supernatant. After 1 h incubation on ice, the samples were centrifuged at 3,000× g for 10 min at 4 °C. The pellet was used for glycogen analysis (see below). For determination of the lipid content, 100 µL of the supernatant were transferred to clean reaction tubes. The tubes were incubated overnight at room temperature to evaporate the solvent. Subsequently, 200 µL of 95% sulfuric acid were added and heated for 10 min at 95 °C in a water bath. After cooling for 5 min, 4.8 mL of phospho-vanillin reagent were added and the tubes were vortexed until development of a reddish color. After 10 min incubation at room temperature, 300 µL of the solution were pipetted into wells of a 96-well plate and measured within 10 min at 535 nm in a plate reader. All samples were measured in triplicate. The content of total lipid was determined from a series of cholesterol standard solutions in concentrations of 100–2,000 µg/mL. The measurements were expressed in µg lipid/µg total protein.

### Heat shock protein 70 analysis

Levels of hsp70 were analyzed by the method described in *Frank et al. (2013)*. Briefly, 20 µg of proteins were separated by discontinuous SDS-Page, transferred to a nitrocellulose

membrane by Western Blot and visualized using monoclonal anti hsp70 antibodies (mouse anti hsp70, antibodies online), a horseradish peroxidase labeled second antibody (goat-anti mouse, DAKO) and detections by 4-chloro-1-naphthol. Gammarids that died during the experiment were excluded from this analysis. To allow inter-gel comparability, a reference sample (fish liver homogenate) was run on all gels. The hsp70-bands, were scanned and quantified densitometrically with ImageJ (*Abràmoff, Magalhães & Ram, 2004*). All sample values were divided by the value of the standard of the respective gel and were expressed as relative hsp70 values.

## Statistical analyses

The weight difference of male and female gammarids was tested with the Welch two sample t-test for unequal variances. A possible effect of the treatment on the mortality rate of gammarids was tested with Kruskal-Wallis-test and Dunn's multiple comparison tests. The effects of gammarid sex, microsporidian and acanthocephalan infection and exposure, as well as the influence of parasitism on host size (separately for females and males) was analyzed by linear models and subsequent ANOVA. Due to the variance heterogeneity, the GLS (generalized least squares fit by restricted maximum likelihood, REML) function of the nlme-library (*Pinheiro et al., 2014*) was used, which allows to correct for unequal variances. Possible correlations of the biochemical parameters were tested with the cor.test-function (Pearson product moment correlation). The statistical analyses were carried out in R v.3.0.1 (*R Core Team, 2013*) and graphs were created in GraphPad Prism v5.0.

# RESULTS

## Prevalence of *P. minutus*

After the exposure experiment, a total of 299 living gammarids was recovered. Among these, 88 of the 160 males (55.0%) and 55 of the 139 females (39.6%) were infected with *P. minutus*. The overall prevalence of this acanthocephalan in the test population was 48% (Table S1).

## Metal analyses

Cadmium levels in control water were lower than 0.1 μg/L during the experiment. In cadmium-exposed groups, concentrations were declining gradually after addition of the stock-solution and reached a stable concentration of 2.89 μg/L on average after about 5 h (Fig. 1). The tissue concentrations of uninfected and *P. minutus*-infected control gammarids were below 0.2 μg/g. Cadmium-exposed female gammarids had almost ten-fold higher cadmium concentrations, and the respective males had close to eight-fold higher concentrations than control animals. Cadmium-exposed female gammarids accumulated cadmium to a higher extent than male gammarids (1.5 and 1.1 times higher for uninfected and *P. minutus*-infected group respectively). The cadmium concentrations determined in *P. minutus* cystacanths from cadmium-exposed gammarids were approximately four times higher than those of cystacanths from control gammarids (Fig. 2).

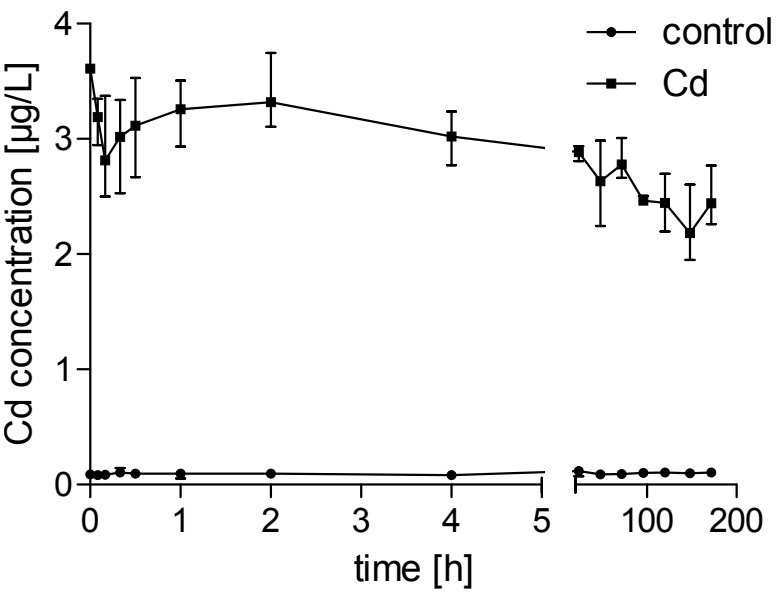

**Figure 1  Water concentrations of cadmium in control and exposure tanks over time.** $n = 3$.

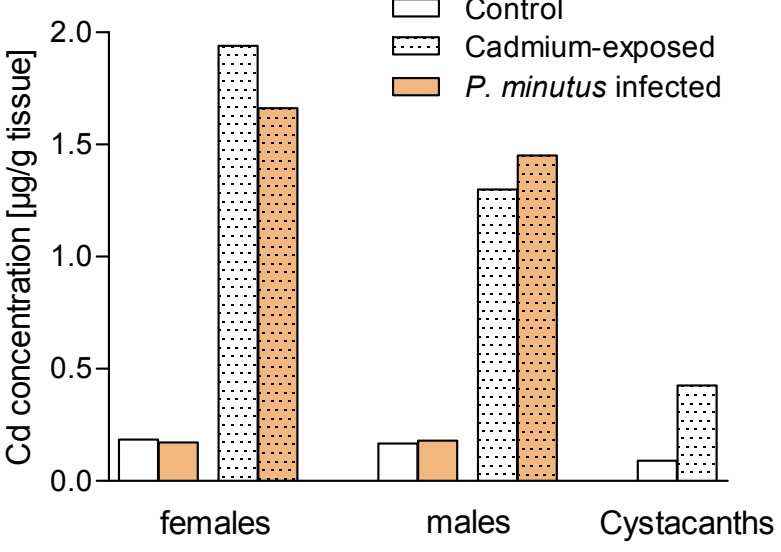

**Figure 2  Cadmium concentrations in tissues.** Cadmium concentrations in pools of *P. minutus* infected and uninfected *G. fossarum* ($n = 2$) and *P. minutus* cystacanths ($n = 1$).

## Microsporidian infections in gammarids

After the exposure experiment, 212 *G. fossarum* individuals were tested for microsporidian infections. Gammarids used for metal analyses could not be tested by PCR, so their microsporidian-infection status remains unclear. By sequencing, three different microsporidian isolates were obtained: (1) 1300 bp (GenBank accession no. KR871381), 99% similarity to *Microsporidium* sp. 505 (FN434085); (2) 433 bp (KR871382), undefined *Microsporidium* sp. MH with 93% identity to *Cystosporogenes* sp. (GQ379704) and (3)

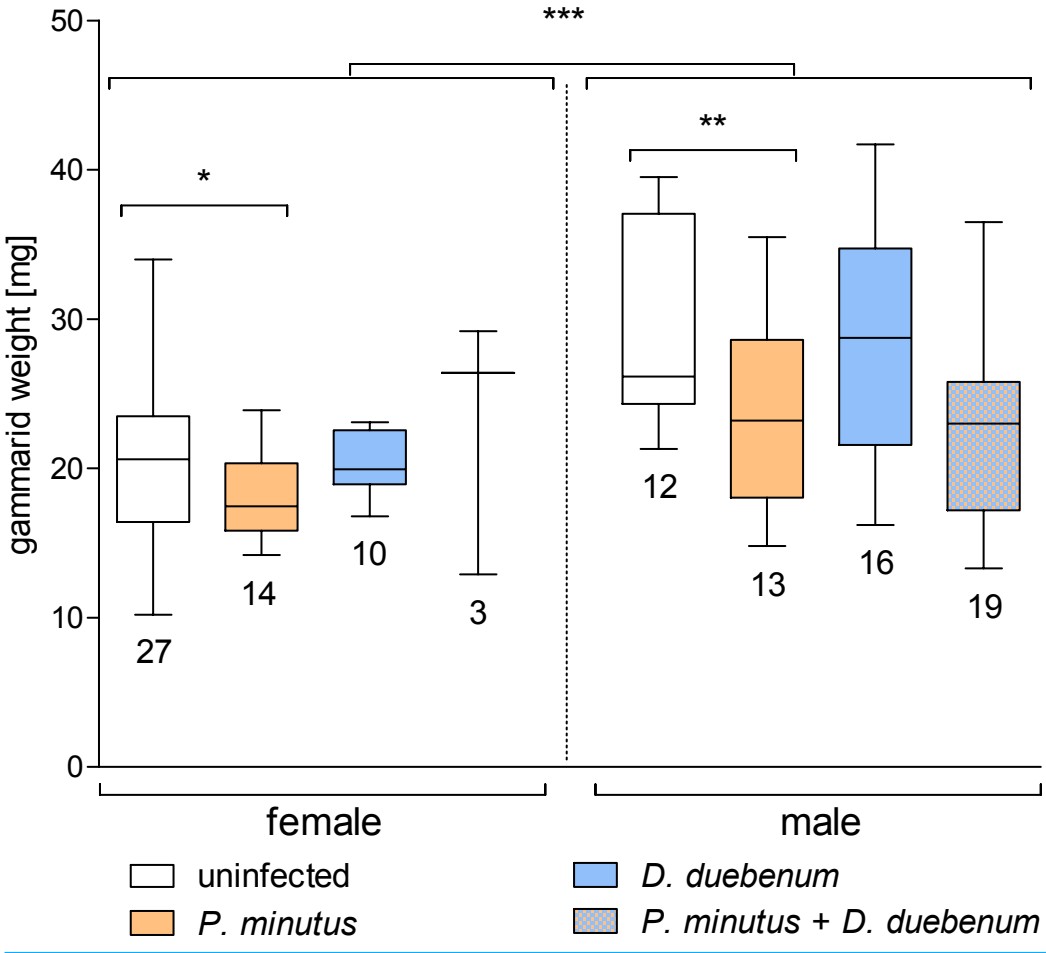

**Figure 3 Box plot of gammarid weight depending on infection and sex.** Numbers below boxes show number of gammarids. Asterisks indicate significant differences.

680 bp (KR871380), 98% similarity to *Dictyocoela duebenum* (JQ673483). Due to the sequence divergence between previous isolates of *D. duebenum* and our *Dictyocoela* species, we can only make a putative identification here. The prevalences of the different microsporidians in the population are shown in Fig. S1. Individuals infected only with *D. duebenum* were considered in biochemical analyses, as this species was the most abundant.

In total, 65% of the gammarids were infected with either one or several microsporidian species simultaneously. The prevalence of microsporidians was higher in males (77.7%) than in females (48.4%). Infections with *D. duebenum* only were recorded in 70.8% of females and 62.3% of males. In total 26.4% of gammarids were infected simultaneously with microsporidians and *P. minutus* cystacanths, with a higher percentage of males (38.0%) compared to females (11.0%) (Table S1).

## Effects of infection on gammarid weight

The mean weight of females (19.96 mg) and males (25.63 mg) differed significantly (Welch two sample $t$-test; $t = -5.18$, $p < 0.001$; (Fig. 3). Additionally, acanthocephalan infected

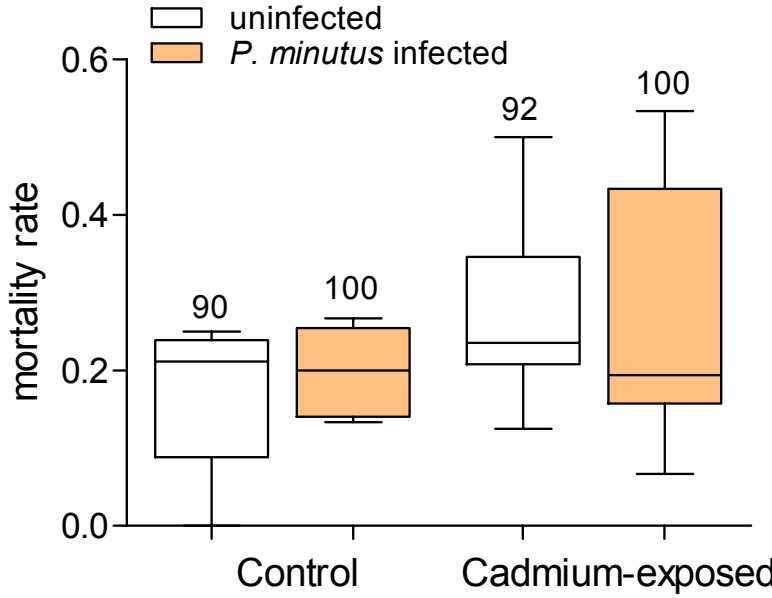

**Figure 4 Mortality rate of *G. fossarum* in the test groups (*n* = 6).** No significant differences between the groups were detected. Numbers above bars indicate number of gammarids.

gammarids had a significantly lower weight than their uninfected conspecifics (females: infected 18.84 mg vs 20.47 mg for uninfected gammarids, $F = 5.90$, $p < 0.05$; males: infected 22.94 mg vs uninfected 28.70 mg for uninfected gammarids, $F = 11.07$, $p < 0.01$). In contrast, microsporidian infection was not correlated with the weight of gammarids (females: $F = 0.01$, $p = 0.92$; males: $F = 0.27$, $p = 0.61$; Fig. 3).

## Mortality rate of gammarids

In the control groups, mortality was slightly higher in the *P. minutus*-infected group (20.0%) compared to the group of uninfected gammarids (17.0%, see Fig. 4). The mortality in cadmium-exposed groups was overall higher, with a slightly higher death rate (27.0%) in uninfected gammarids compared to *P. minutus*-infected ones (26.1%). On average, the mortality in the control group was 18.4%, whereas 26.6% of the cadmium exposed gammarids died. No significant differences between the groups were detected (Fig. 4).

## Biochemical analyses

Uninfected, *P. minutus* infected, *D. duebenum* infected and gammarids infected with both parasites were used for further analyses. At all a total of 54 female and 60 male gammarids was available for the biochemical analysis. The output of the GLS model is shown in Table S2.

### *Glycogen*

Overall, cadmium exposure decreased glycogen levels, but the analyses indicated a significant increasing effect of cadmium-exposure on glycogen levels in microsporidian infected female gammarids. A similar increase was found in unexposed males (Table 2 and Fig. 5).

**Table 2** **Analysis.** Results of the ANOVA of the different groups on biochemical parameters.

| Group | DF | F-value | p-value | DF | F-value | p-value | DF | F-value | p-value |
|---|---|---|---|---|---|---|---|---|---|
| | | **Glycogen** | | | **Lipid** | | | **hsp70** | |
| (Intercept) | 1 | 10475360,20 | 0,000 | 1 | 20852,74 | 0,000 | 1 | 772,39 | 0,000 |
| sex | 1 | 52,96 | 0,000*** | 1 | 82,41 | 0,000*** | 1 | 2,63 | 0,108 |
| expo | 1 | 19,58 | 0,000*** | 1 | 6,30 | 0,014* | 1 | 13,40 | 0,000*** |
| micro | 1 | 39,65 | 0,000*** | 1 | 23,05 | 0,000*** | 1 | 2,18 | 0,143 |
| acantho | 1 | 0,39 | 0,535 | 1 | 10,31 | 0,002** | 1 | 0,98 | 0,325 |
| sex:expo | 1 | 0,09 | 0,768 | 1 | 0,00 | 0,986 | 1 | 6,78 | 0,011* |
| sex:micro | 1 | 0,00 | 0,996 | 1 | 4,14 | 0,045* | 1 | 0,01 | 0,938 |
| expo:micro | 1 | 5,90 | 0,017* | 1 | 3,39 | 0,068 | 1 | 13,06 | 0,000*** |
| sex:acantho | 1 | 4,40 | 0,039* | 1 | 10,92 | 0,001** | 1 | 2,48 | 0,119 |
| expo:acantho | 1 | 0,16 | 0,690 | 1 | 4,31 | 0,041* | 1 | 3,76 | 0,055 |
| micro:acantho | 1 | 4,07 | 0,046* | 1 | 2,38 | 0,126 | 1 | 0,58 | 0,448 |
| sex:expo:micro | 1 | 8,56 | 0,004** | 1 | 0,00 | 0,987 | 1 | 0,03 | 0,855 |
| sex:expo:acantho | 1 | 1,47 | 0,228 | 1 | 0,35 | 0,556 | 1 | 0,79 | 0,376 |
| sex:micro:acantho | 1 | 1,21 | 0,273 | 1 | 0,24 | 0,624 | 1 | 1,19 | 0,277 |
| expo:micro:acantho | 1 | 0,15 | 0,699 | 1 | 0,41 | 0,523 | 1 | 2,28 | 0,134 |
| sex:expo:micro:acantho | 1 | 0,62 | 0,434 | 1 | 5,28 | 0,024* | 1 | 0,48 | 0,489 |

**Notes.**
sex, host sex; expo, exposure condition; micro, microsporidian infection; acantho, acanthocephalan infection; DF, degrees of freedom.

Also, a significant interaction was present for microsporidian and acanthocephalan infection showing increased glycogen levels (Table 2, Fig. 5 and Table S2). According to the GLS, increased glycogen levels are also present in females compared to males, independent of exposure and parasites (Table 2, Fig. 5 and Table S2).

## Lipids

Similar to glycogen, total lipids were significantly decreased by cadmium exposure, but this effect was modulated by parasite infection. Also, the total lipid content was highly dependent of host sex, with lower concentrations in males than in females (Fig. 6 and Table 2). The acanthocephalan infection significantly decreased the total lipid level in all groups, except for cadmium-exposed males, which is reflected by the significant interactive effect of host sex and *P. minutus* infection (Table 2 and Table S2). Microsporidians caused a significant increase of total lipids especially in females. This increase was less pronounced in cadmium exposed individuals (Fig. 6 and Table 2).

### hsp70 levels

For 3 samples (3 exposed, uninfected females), no hsp70 values were obtained (no visible band) and 2 (1 exposed and 1 unexposed, uninfected female) dropped out as outliers (>3 fold above group average). The combination of cadmium exposure and microsporidian infection had a significant increasing effect on hsp70 levels, while microsporidians alone had no effect on hsp70 (Table 2, Fig. 7 and Table S2). Cadmium exposure also increased the hsp70 level in the other infected and uninfected groups. Infection with *P. minutus* did not have a significant influence on the hsp70 levels (Table 2 and Table S2). Nevertheless, a slight

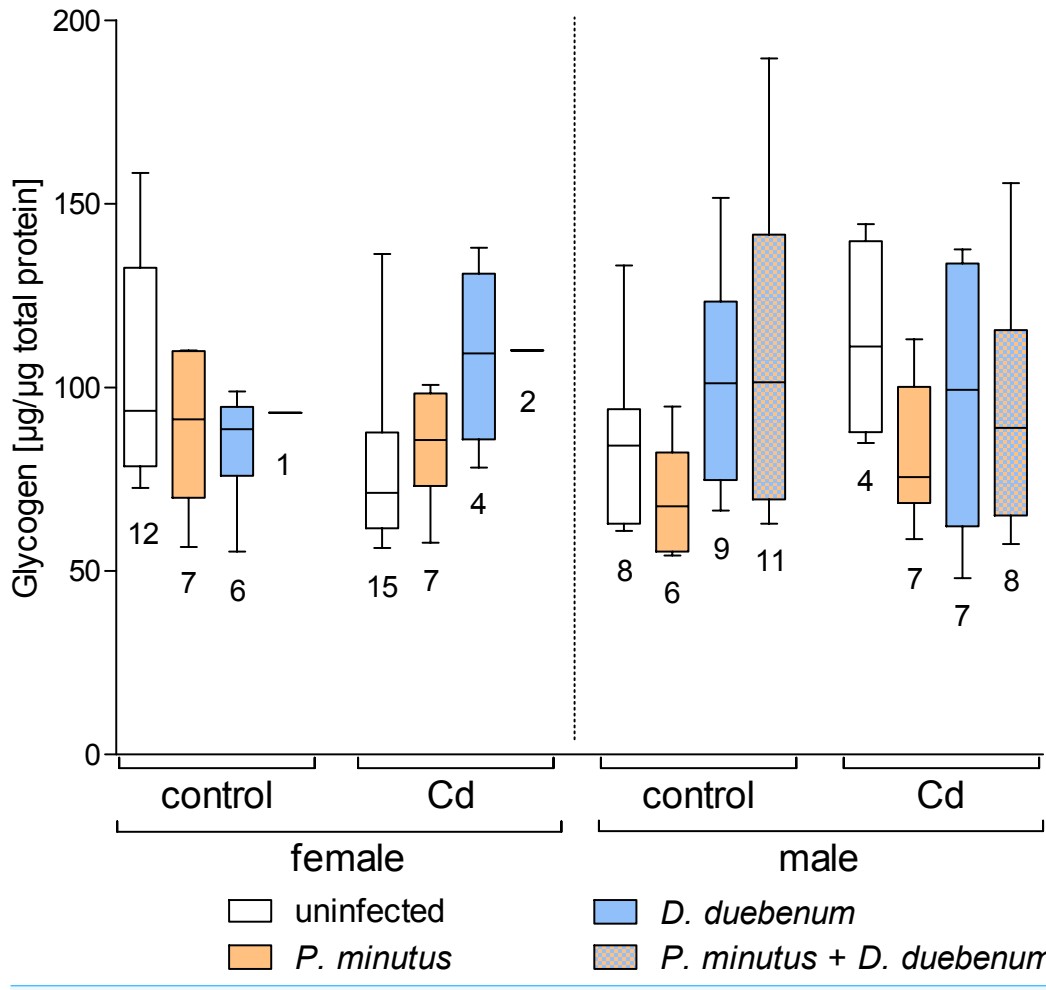

**Figure 5 Glycogen content in infected and uninfected *G. fossarum* females and males after cadmium exposure.** Numbers below boxes show number of gammarids. For significant differences, see Table 2.

increase in the hsp70 level can be observe for unexposed and acanthocephalan infected females according to Fig. 7.

### Correlations

Correlation analyses revealed significant negative correlation between lipids and hsp70 ($n = 109$, $p < 0.001$, $\rho = -0.32$), and weak positive correlations between lipid and glycogen ($n = 109$, $p < 0.01$, $\rho = 0.28$). No significant correlation was found for hsp70 and glycogen.

## DISCUSSION

The present study provides data on combined and single effects of parasite infections (Acanthocephala and Microsporidia) and cadmium-exposure on mortality, energy reserves of the gammarid host and its stress response quantified by the hsp70 levels. Three microsporidian isolates were detected in the naturally infected *Gammarus* population, including one novel isolate. For the biochemical analyses, only infections

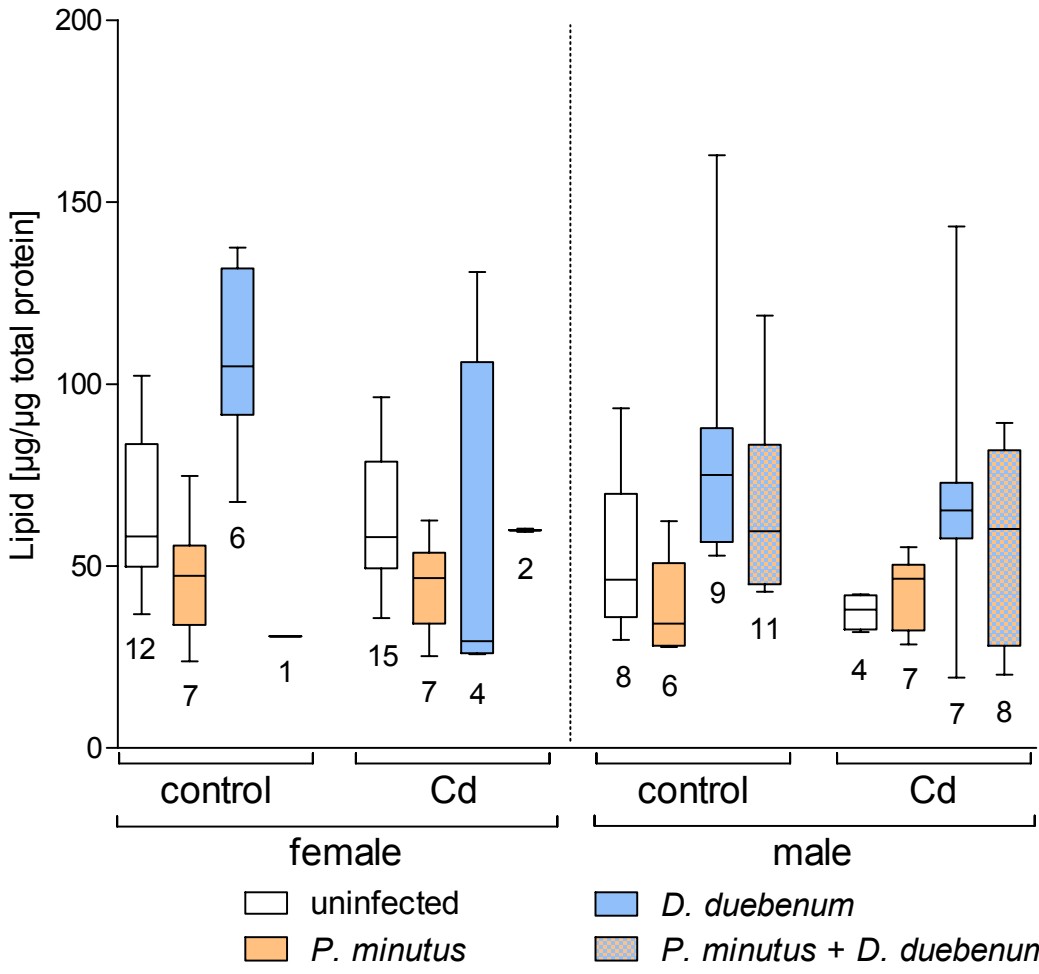

**Figure 6 Lipid content in infected and uninfected *G. fossarum* females and males after cadmium exposure.** Numbers below boxes show number of gammarids. For significant differences, see Table 2.

with *D. duebenum* (species assignment putative) were considered. The latter parasite was found frequently in different amphipod species and is characterized by a vertical mode of transmission and is known to feminize the host population by sexual conversion (*Terry et al., 2004*). Virulence of this microsporidian is considered to be low, but sub-lethal effects on the host were not well studied until now. Surprisingly, we did not observe a female bias among *Dictyocoela*-infected *G. fossarum* in the present study. Nevertheless, this parasite might influence host metabolism in the attempt to elicit feminization. This might have influenced the outcome of our analyses, but this effect remains speculative here.

According to our results, the weight of *G. fossarum* was lower in acanthocephalan infected male and female hosts, while microsporidian infection showed no significant effect. This indicates that either smaller individuals were more likely to be infected with *P. minutus*, or that this parasite influences host feeding behavior and its growth rates (*Agatz & Brown, 2014*).

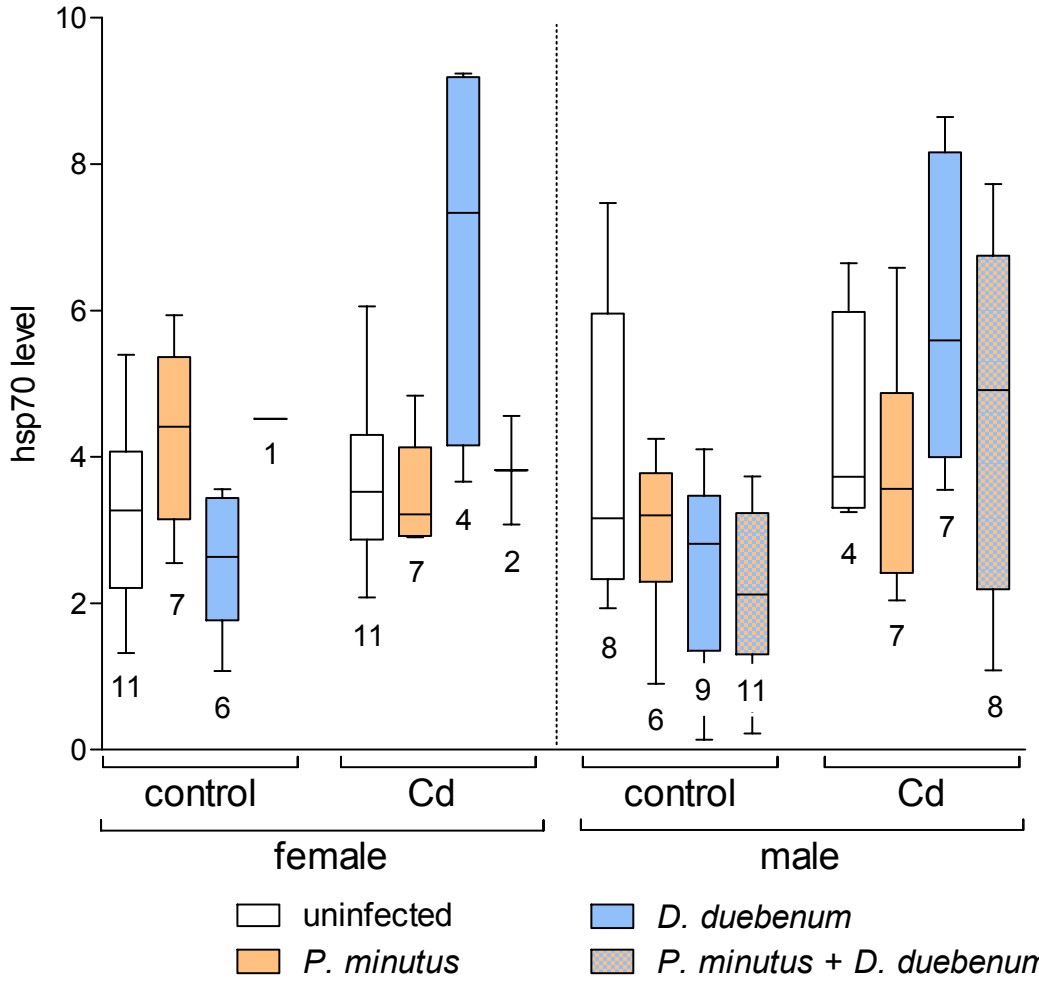

**Figure 7 Relative hsp70 content in infected and uninfected *G. fossarum* females and males after cadmium exposure.** Numbers below boxes show number of gammarids. For significant differences, see Table 2.

The cadmium concentration used in the present study was about factor 100 below the 96h LC50 values reported for *G. fossarum* in previous studies (*Alonso, De Lange & Peeters, 2010*; *Boets et al., 2012*), therefore no severe mortality was expected. Nevertheless, besides a baseline mortality of about 20% we observed an additional mortality of 6%–10% in the cadmium exposed groups. Our results showed also no negative effect of *P. minutus* on host mortality, while mortality was slightly higher (although not significant) in cadmium exposed *G. fossarum*. *Gismondi, Cossu-Leguille & Beisel (2012b)* found an increased cadmium tolerance in male *G. roeseli* infected with *P. minutus*, while infected females became more susceptible to combined stressors. Such sex dependent difference in susceptibility to cadmium was also reported for *G. fossarum*, where females were less tolerant to the exposure than males (*Schill, Görlitz & Köhler, 2003*).

The energy reserves of *G. fossarum* were influenced by both cadmium-exposure and parasite infection. Glycogen content increased in *D. duebenum* infected individuals with

a heterogeneous effect of host sex. *Gismondi et al. (2012a)* also reported an increase in glycogen in *Dictyocoela*-infected *G. roeseli*, but a decrease after cadmium exposure. In another study (*Gismondi et al., 2012b*), an increase of glycogen levels was detected depending on both microsporidian infection and cadmium exposure, similar to the present study. Only females infected with microsporidians were used in the latter study, therefore effects of host sex cannot be compared. In our study, slightly elevated glycogen levels in acanthocephalan infected female amphipods were found. This effect due to acanthocephalan infection was also described previously (*Plaistow, Troussard & Cézilly, 2001*), but the sex effect was dependent on the season when amphipods were sampled (*Gismondi, Beisel & Cossu-Leguille, 2012b*). An increase in glycogen levels in acanthocephalan infected individuals seems to be a common phenomenon. It might be explained by the increased uptake of nutrients due to extended energy requirements of the infected host or due to a quick mobilization of energy storage during the growth phase of the parasite (*Sparkes, Keogh & Pary, 1996*). In contrast, the need for detoxification of pollutants might cause an increased metabolic activity that depletes glycogen storages. Furthermore, presence of both acanthocephalan and microsporidian infection caused significantly increased glycogen concentrations. This is in contrast to *Gismondi et al. (2012b)* who found depleted glycogen levels in *G. roeseli* infected with microsporidians and acanthocephalans at the same time.

Similar to glycogen, an increase of total lipids was observed in males and females of *G. fossarum* infected with the microsporidium *D. duebenum*, while cadmium-exposure had a decreasing effect. Previously, a similar increase in lipid levels due to microsporidians compared to the uninfected control was found for *G. roeseli* exposed to a nominal cadmium concentration of 8 µg/L (*Gismondi et al., 2012b*), but not at 2 µg/L or in unexposed control groups. Therefore, the nominal cadmium concentration of 4 µg/L (2.9 µg/L measured) used in the present study might be the threshold that caused an effect in the exposed amphipods. Interestingly, the opposite was observed in *G. roeseli* infected with *P. minutus*, where decreased lipid levels were reported after exposure to 2 µg/L cadmium concentration, but not to 8 µg/L (*Gismondi et al., 2012b*; *Gismondi, Beisel & Cossu-Leguille, 2012b*). In the present study, the acanthocephalan infected group showed slightly decreased lipid levels, which was most pronounced in females independent of exposure condition. Similar to glycogen storage, the parasite might drive host metabolism to utilize the lipid storage depending on its own requirements. The variable results observed for the lipid content in different studies could be due to seasonal variation in reserves substances like lipids or glycogen (*Gismondi, Beisel & Cossu-Leguille, 2012a*; *Gismondi, Beisel & Cossu-Leguille, 2012b*). Additionally, they could indicate different response patterns to cadmium and parasites by the two host species *G. fossarum* and *G. roeseli*.

To summarize, the response of *G. fossarum* and *G. roeseli* to cadmium and parasites on the level of energy reserves is essentially similar although slight species-specific differences might be present. Due to the positive correlation between glycogen and lipid levels in the present study, it can be assumed that they respond similarly to parasite infection. Especially lipids represent nutrient reserves to prevent starvation and to provide energy for

reproduction; therefore, changes of lipid levels represent long-term physiological effects in the host that might impair its survival and that can have an effect on the population development under stressed conditions (*Cargill et al., 1985*).

Increased levels of hsp70 indicate a general stress response (protein damage) of an organism, e.g., to chemicals or effects of parasite infections (*Frank et al., 2013*). In the present study, a clear stress-effect was detected in microsporidian-infected *G. fossarum* exposed to cadmium, while no significant effect of infection with *P. minutus* was observed. A similar increase of hsp70 due to multiple microsporidian infection was already observed previously in *G. pulex* (*Grabner, Schertzinger & Sures, 2014*). In the study of *Gismondi et al. (2012b)*, malondialdehyde as a marker for lipid oxidation and therefore cellular damage, similar to hsp70, was increased in *G. roeseli* after cadmium-exposure. This effect was even more pronounced, if the amphipods were additionally infected with microsporidians, acanthocephalans or with both together. These findings provide evidence for the idea that intracellular replication of microsporidians causes a destabilization of cell metabolism leading to protein damage (measured by hsp70 or malondialdehyde), if additional stressors (here cadmium) are present.

An irregular pattern of the hsp70 response was observed in previous studies, caused by the combined effect of *P. minutus* infection and thermal or chemical stress on gammarids. No increased levels of hsp70 were detected in infected *G. roeseli* after exposure to either heat or palladium (*Sures & Radszuweit, 2007*). Similar results were reported for *G. fossarum* infected with *P. minutus*, comparing uninfected and cadmium-exposed animals (*Frank et al., 2013*). Such response patterns were not observed in the present study, as *P. minutus* induced no considerable hsp70 response. Again differences between the amphipod species might provide an explanation. *Peschke et al. (2014)* found remarkable differences in the hsp70 response of uninfected *G. pulex* and *G. roeseli* populations from sampling sites with different pollutant burdens. The response of hsp70 in *G. pulex* showed variations, while no effect was seen in *G. roeseli* from the same sites. Similar differences might exist between *G. roeseli* and *G. fossarum*. Nevertheless the different findings of *Frank et al. (2013)* and the present study, both using *G. fossarum*, remain to be elucidated. Interestingly, a negative correlation was found between hsp70 and lipids, which indicates that individuals with depleted energy reserves are more prone to stress effects.

In the present study cadmium-concentrations were slightly lower in males of *G. fossarum* than in females. No significant differences were detected if *P. minutus*-infected and uninfected amphipods were compared. In contrast, significant lower cadmium-concentrations were detected in *G. roeseli* (*Gismondi, Cossu-Leguille & Beisel, 2012b*) or *G. fossarum* (*Frank et al., 2013*) infected with *P. minutus* although both studies used similar sublethal concentrations of cadmium. This might be explained by the low number of individuals analyzed in the present study that did not allow detection of such differences. Cystacanths showed a lower cadmium accumulation potential compared to host tissues, but this finding is based only on one replicate. Similar findings were described before for other host-parasite systems, e.g., *Asellus aquaticus—Acanthocephalus lucii*

(*Sures & Taraschewski, 1995*), *G. pulex—Pomphorhynchus laevis* (*Siddall & Sures, 1998*) as well as for *G. fossarum—P. minutus* (*Frank et al., 2013*).

## CONCLUSION

Microsporidian infections were found to influence the metabolism and the stress response of *G. fossarum* exposed to sub-lethal cadmium concentrations. In contrast, the effect of *P. minutus* cystacanths on energy reserves and stress response of the host was less pronounced. Our data together with data from previous studies indicates slight species dependent differences in the response to multiple stressors (e.g., parasitism and metal pollution). Furthermore, we highly recommend that parasitism should be taken into account in future ecotoxicological surveys, as various aquatic organisms are commonly infected and monitoring data could be misinterpreted if their infection status is not considered.

## ACKNOWLEDGEMENT

We thank Stephen Short and another anonymous reviewer for the helpful and constructive review that helped to improve the manuscript.

### Funding

The German Academic Exchange Service (DAAD) and the Ministry of Science and Technology, Taiwan funded the research stay of Ms. Hui-Yu Chen. The funders had no role in study design, data collection and analysis, decision to publish, or preparation of the manuscript.

### Grant Disclosures

The following grant information was disclosed by the authors:
German Academic Exchange Service (DAAD).
Ministry of Science and Technology.

### Competing Interests

The authors declare there are no competing interests.

### Author Contributions

- Hui-Yu Chen performed the experiments, analyzed the data, wrote the paper, prepared figures and/or tables, reviewed drafts of the paper.
- Daniel S. Grabner conceived and designed the experiments, performed the experiments, analyzed the data, wrote the paper, prepared figures and/or tables, reviewed drafts of the paper.
- Milen Nachev performed the experiments, analyzed the data, wrote the paper, reviewed drafts of the paper.
- Hsiu-Hui Shih conceived and designed the experiments, wrote the paper, reviewed drafts of the paper.

- Bernd Sures conceived and designed the experiments, contributed reagents/materials/analysis tools, wrote the paper, reviewed drafts of the paper.

## DNA Deposition

The following information was supplied regarding the deposition of DNA sequences:
GenBank accession numbers:
KR871381, KR871382, KR871380

## Supplemental Information

Supplemental information for this article can be found online at http://dx.doi.org/10.7717/peerj.1353#supplemental-information.

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
