# Peer review of "Effects of the acanthocephalan Polymorphus minutus and the microsporidian Dictyocoela duebenum on energy reserves and stress response of cadmium exposed Gammarus fossarum"

_PeerJ, doi:10.7717/peerj.1353_

## Round 0.1 · original submission · Major Revisions

Dear Authors,

Thank you for submitting your manuscript to PeerJ. Please note that both reviewers found merit in your study but have recommended a number of issues that need to be addressed prior to acceptance.

Reviewer 1 ·

Basic reporting

"No Comments"

Experimental design

"No Comments"

Validity of the findings

"No Comments"

Additional comments

The present manuscript is an interesting experimental work about the effect of two groups of parasites on energy reserves in gamamrids exposed to a toxic stress. The study highlights the importance of studying the parasitism of the individuals in ecotoxicological studies. Overall, I think that it is a nice piece of research and definitely has potential for this journal. However, some parts need some improvements before publication.
I have tried to summarize the major issues that should be reconsidered by the authors:
Abstract
Line 33: Please reword the sentence, because according to the material and methods, metal analyses were not performed in dead animals.
Line 38-39: This result is not discussed in the manuscript.
Material and methods
Line 116: Please include a reference.
Lines 116-117: The sorting of gammarids under a binocular was done to know its acanthocephala infection status? was the number of cystacanths counted in each gammarid?
Line 125: Why authors choose an exposure time of 7 days? Is this time enough to study differences in the energy reserves? I think that this is a key issue that must be clearly explained.
Line 126. Please, specify whether once in the lab, the gammarids selected for the exposure experiment were fed or kept in fasting conditions before the experiment. In this sense, was food provided during the exposure experiment? Similarly, because cannibalism is common in gammarids, I think that is important to know if dead animals were checked daily and if these were left or removed from the tanks. I think that this information is essential to properly understand the data on energy reserves.
Lines 129-131 and 140: I found an inconsistency regarding the methodology used in the metal analysis in water samples. In lines 129-131 authors stated that samples were filtered and acidified, while in line 140 authors stated that samples were analysed without any pretreatment. Please check and reword it to clarify the methodology followed.
Line 154: Due to the number of gammarids used in each analysis differed, I think it would be helpful to specify the number of individuals used in each analysis. Similarly, I think that is important to clearly note that for some analyses (weight, lypids, …) only gammarids infected with one species of microsporidian were used (e.g., lines 33, 275 or 295). Additionally, authors could provide information about how many gammarids were used to obtain each pool of cystacanths, and if more than one cystacanths was obtained per gammarid.
Line 222: Were dead gammarids used in any analysis? if done, please specify in which were included
Results
Overall the result section is a bit difficult to follow. A higher effort should be carried out to improve the clarity of the results. For example, authors addressed statistical analyses with several factors and interactions (Table 4), being the effects of some of the factors and their interactions (at least the most relevant, that I think that would be those related to Cd exposure; e.g., lines 299-300; lines 308-309) not properly described in the text, and they can be not easily extracted from the figures. Additionally, I think that could be useful to maintain the same scheme that is followed in material and methods.
Lines 243-244 and 261-262: Could be sex or microsporidian infection identified in dead gamamrids?
Line 247: Please clarify why the analysis of metals in gammarids was based on only two replicates of two gammarids each and on only one replicate for cystacanths, when more than 300 gammarids were used in the experiments.
Line 275: What about the effect of Cd in the body weight?
Line 285: I think that the sentence can be deleted or reworded because it relates to the experimental design, not result, and can lead to confusion.
Line 286: The mortality of the control group was higher than mortalities recommended for acceptability of control groups in toxicity test (10%; e.g., ASTM, 1999). What do authors think about the high mortality detected in control groups? In addition, authors should be cautious because a concentration that causes mortalities of 26-27%, is high enough to be considered as lethal and not sub-lethal.
Lines 327-328: What about the correlation between glycogen and hsp70? Please include the number of data used for each correlation. Please check the statistic parameters; a R2 can not be negative!
Discussion
Line 346: Please use references related to G. fossarum (e.g., Alonso et al., 2010; Boets et al., 2012; Musko et al., 1990), none of the cited references studied the effect of cadmium on this species.
Lines 357-359 and 364-367: Both sentences seem to say the same. Please reword these to avoid duplicate information.
Lines 357-372: According to the result section, your data showed higher levels of glycogen in microsporidian-infected and cadmium-exposed females as well as in unexposed males. However, in the discussion authors said that “The effect of cadmium exposure on glycogen in G. fossarum in the present study was similar to previous results for G. roeseli. Cadmium caused an overall decrease of glycogen levels, but microsporidian-infected individuals showed increased values…”. I think that this lines are confusing, and therefore I suggest reword it to improve its comprehensibility and coherence.
Lines 436-437: I think that “as well as” must be replaced by something as “together to” because the present manuscript only addresses the study of one species. Thus, I think that is not possible to state “our data does not indicate species dependent differences …”
Overall, due to the effective concentration for most of the experimental periods were the 2.9 µg Cd/L, I think that could be more useful for readers if the effective concentration instead of the nominal concentration was stated through the text (e.g., Table 3, and lines 377-378).
Tables and figures
The manuscript in the current version has 4 tables and 7 figures, plus 2 tables and 1 figure additional in the supplemental material. Due to not variation was detected in the three parameters checked in water during the experiment, I think that Table 1 could be deleted, and water information integrated in the text.
Both Table 3 and Figure 4 show the percentages of mortality of infected and uninfected gammarids from different treatments after the Cd experiment. Thus, in order to avoid duplicate information, I suggest deleting the figure, and completing the table with the additional information contained in the figure such as min, max, SD…
Figure 2: I think that the bars of cystacanths are not correct and both would be in orange. Now the bars are indicating cystacanths uninfected (?) and cystacanths exposed to Cd. Overall, I think the legends of the different data series included in the figure are a bit confusing, so, I suggest redo the figure.
Reference
Alonso et al. 2010. Contrasting sensitivities to toxicants of the freshwater amphipods Gammarus pulex and G. fossarum. Ecotoxicology 19: 133-140.
ASTM 1999. Standard Guide for Conducting 10-day Static Sediment Toxicity Tests with Marine and Estuarine Amphipods. E 1367-99.
Boets et al., 2012. A comparison of the short-term toxicity of cadmium to indigenous and alien gammarid species. Ecotoxicology 21: 1135-1144.
Musko et al., 1990. The impact of Cd and different pH on the amphipod Gammarus fossarum Koch (Crustacea: Amphipoda). Comp Biochem Physiol C 96:11–16.

·

Basic reporting

The article is generally well written, follows a suitable format and the raw data has been made available. The article should has sufficient introduction and background but I do have following points:

The experiment compares the response of infected and uninfected animals to cadmium exposure. However, although the rationale underlying other aspects of the experiment, no reasoning is given for the choice of cadmium as the exposure metal (although I can guess why it was chosen). In fact cadmium is not mentioned in the introduction at all. It may be worth explicitly detailing why this metal was used. Also, no rationale is given for the choice of concentration. It is mentioned in the discussion that 4 ug/L is a sub-lethal concentration, a factor of 10 below the LC50 concentration. Is this concentration environmentally relevant? Given that the study uses a single metal at a single concentration, some justification of the choice should probably be made, even if the metal and concentration was selected to simply to allow comparison of the author’s data with previous studies.

line 86: '…parasites for amphipods (Haine et al., 2004; Grabner, Schertzinger & Sures, 2014) (Grabner et al., 2015). '
Why are the Haine et al., 2004; Grabner, Schertzinger & Sures, 2014 and Grabner et al., 2015 references in separate parentheses?

line 97-99: 'Additionally, heat shock proteins (hsp) can be used as general stress markers, suitable to detect effects of parasites and metal exposure (Schill, Görlitz & Köhler, 2003; Frank et al., 2013).
'
Is hsp70 used to monitor stress in crutaceans? These references may contain this information but if hsp70 is a efficient marker in crustaceans, it may be useful for the authors to make this clear.

line 101-105: Both G. roeseli and G. fossarum are suitable hosts for P. minutus, however it was found that they respond differently (e.g. on biochemical and behavioral level) to infection (Bauer et al., 2000; Helluy, 2013). We assumed that this species might also react differently on the combination of stressors such as metal pollution and parasite infections.
In the second sentence the ‘this’ in the ‘…we assumed that this species might..’ probably should name the species in question.
also ‘might also react differently on the combination of stressors’ should be: differently to the combination of stressors.

Experimental design

Line 113-116: 'Gammarids were collected by kick-sampling from a small stream (Rumbach) near Mülheim (N51° 25' 28.628"; E6° 54' 25.553"), NRW, Germany in March 2014. This site is characterized by a mixed population of Gammarus pulex and Gammarus fossarum, and high prevalence of the acanthocephalan Polymorphus minutus in both amphipod species.
'
The authors describe the site as ‘characterized’. Has this site been studied previously? If so, then a reference could be added.
116-118: ‘After sorting the collected gammarids under a binocular a total of 382 G. fossarum was used for the experiment. The gammarids were grouped by acanthocephalan infection’

I assume the an animals are designated as infected due to the appearance of cystacanths within the body of the amphipod. I have several queries:
Do the cysts clearly contain an acanthocephalan parasite? Cysts can form around many parasite species. For example, other acanthocephalan species and trematode parasites. Are the authors confident that the cysts contain Polymorphus minutus? For the later microsporidian identification, the authors use molecular techniques and sequence the small ribosomal subunit. Has this been done for the acanthocephalan species infecting G fossarum at this site? Also, assuming the cysts contain Polymorphus minutus, can the authors determine if a cyst contains a living parsite? The melanisation and encystment by the host can kill the parasite (this from personal experience of examining trematode parasites). Of course, dead or dying parasite may have less of an influence on the host’s metabolism.

Validity of the findings

Some of the comparisons are made using relatively few gammarids (e.g. there are four animals in the female, microsporidian infected, cadmium exposed group). These low sample numbers may have consequences. For example, amphipods undergo moult cycles and females go through striking reproductive cycles. Low sample sizes may fail to average out the influences of these cycles. Furthermore, it is reasonable to suggest that these moult and reproductive cycles may have strong influences on glycogen and lipid levels. Having said this, the authors been clear about how many animals are in each group and detail the statistical tests applied. Readers of PeerJ will be able to make up there own minds about the samples sizes for some of the comparisons.
Line 265-268: ..with 93% identity to Cystosporogenes sp. (GQ379704) and (3) 680 bp (KR871380), 98% similarity to Dictyocoela duebenum (JQ673483). The prevalences of the different microsporidians in the population are shown in supplemental figure S1. Individuals infected only with D. duebenum were considered in biochemical analyses, as this species was the most abundant.

The authors identify the most abundant microspoiridan as D. duebenum on the basis of the 98% identity with JQ673483. This is reasonable. However, it is possible others may issue. The JQ673483 sequence (described as Dictyocoela duebenum) itself differs from the reference strain of D duebenum (AF397404) by 1.4%. The extent of sequence difference before a given parasite is considered a novel Dictyocoela species but not Dictyocoela duebenum is an open question. An arbitrary difference of 1% has been suggested (ref). It may be worth stating that describing the identification of the microsporidian Dictyocoela duebenum as putative at this point.

line 270: Finding higher level of infection in males than females is of interest, I assume this holds for just the Dictyocoela duebenum infection. In fact, it may be worth giving the infection levels for just Dictyocoela duebenum in addition to microspridian infection as a whole. A statisically higher infection prevalence in females is used as evidence for successful sexual conversion (Terry et al., 2004). The infection distribution observed is inconsistent with this Dictyocoela duebenum being a successful feminser in G fossarum, as the reference strain (AF397404) is in Gammarus duebeni. The microsporidian may well attempt to feminise this host and make other metabolic demands but the infection prevalences suggest that can not efficiently convert males in to females in this case. This is probably be worth mentioning in the discussion.

---

## Round 0.2 · Minor Revisions

Dear Authors,

Thank you for submitting your revised manuscript. Both reviewers felt their comments have been fully addressed. Please see additional comments from reviewer 1 which you may wish to consider before the final acceptance.

Reviewer 1 ·

Basic reporting

No comments

Experimental design

No comments

Validity of the findings

No comments

Additional comments

The authors did a good job addressing most of my previous comments. However, some questions emerged from the revision and they should be considered in a new version of the manuscript:
To improve the manuscript readability, I think that the text structure used in material and methods and results sections should be maintained in the discussion.
Lines 441-445: This paragraph must be in the context of Cd mortality; what are the authors referring with secondary infections?
Line 445: Please check the data reported in the references cited in relation to the 96h LC50; 10-factor or 100-factor?.
Line 485: Nominal concentrations are sometimes used when nominal and measured concentration would be similar. Here, the measured concentration is 27% lower than the nominal one, and therefore, I think that the use through the text of real or measured concentration is more appropriate.
Lines 544-545: The absence of differences could be related to the small sample size in the analysis of metals: it was based on only two replicates of two individuals each with gammarids and on only one replicate with cystacanths. This fact should be clearly stated in the text.

·

Basic reporting

All my concerns have been answered.

Experimental design

All my concerns have been answered.

Validity of the findings

All my concerns have been answered.

---

## Round 0.3 · accepted · Accept

Dear Authors,

Thank you for submitting a revised version of your manuscript. I am please to inform you that your paper has been accepted for publication in PeerJ.